# In Situ Proteolysis Condition-Induced Crystallization of the XcpVWX Complex in Different Lattices

**DOI:** 10.3390/ijms21010308

**Published:** 2020-01-02

**Authors:** Yichen Zhang, Shu Wang, Zongchao Jia

**Affiliations:** 1Department of Biomedical and Molecular Sciences, Queen’s University, 18 Stuart Street, Kingston, ON K7L 3N6, Canada; 11yz53@queensu.ca; 2College of Chemistry, Beijing Normal University, 19 Xinjiekou Outer Street, Beijing 100875, China; piggy@mail.bnu.edu.cn

**Keywords:** in situ proteolysis, X-ray crystallography, chymotrypsin digestion, type II secretion system, pseudopilin tip complex

## Abstract

Although prevalent in the determination of protein structures; crystallography always has the bottleneck of obtaining high-quality protein crystals for characterizing a wide range of proteins; especially large protein complexes. Stable fragments or domains of proteins are more readily to crystallize; which prompts the use of in situ proteolysis to remove flexible or unstable structures for improving crystallization and crystal quality. In this work; we investigated the effects of in situ proteolysis by chymotrypsin on the crystallization of the XcpVWX complex from the Type II secretion system of *Pseudomonas aeruginosa*. Different proteolysis conditions were found to result in two distinct lattices in the same crystallization solution. With a shorter chymotrypsin digestion at a lower concentration; the crystals exhibited a P3 hexagonal lattice that accommodates three complex molecules in one asymmetric unit. By contrast; a longer digestion with chymotrypsin of a 10-fold higher concentration facilitated the formation of a compact P2_1_2_1_2_1_ orthorhombic lattice with only one complex molecule in each asymmetric unit. The molecules in the hexagonal lattice have shown high atomic displacement parameter values compared with the ones in the orthorhombic lattice. Taken together; our results clearly demonstrate that different proteolysis conditions can result in the generation of distinct lattices in the same crystallization solution; which can be exploited in order to obtain different crystal forms of a better quality

## 1. Introduction

For decades, X-ray crystallography has been the most powerful and robust approach to determining atomic protein structures, which provides incisive insights into the three-dimensional spatial arrangements of protein structures and allows for an in-depth exploration and understanding of the structure-related protein functions [1,2,3,4,5]. A majority of the established protein structures have been determined by X-ray crystallography, because it is a well-established methodology for solving protein structures with a large span of molecular weights [2,6,7]. To obtain crystal structures, it is required to experimentally obtain high-quality protein crystals for the X-ray diffraction experiments, which is proven to be the bottleneck for many proteins [8,9], especially for those large proteins or protein complexes that are over 100 kDa [10,11,12]. It is estimated that of all of the proteins used for crystallization trials, only one-third can be crystallized, of which about a half can further produce usable-quality X-ray diffraction data through devising and diversifying the proper optimization of crystallization conditions (http://targetdb.pdb.org) [13,14].

Years of protein crystallographic studies have revealed that stable fragments or domains of proteins are more prone to crystallization, which increases the possibility of forming high-quality crystals for diffraction experiments [15,16]. Stable molecules tend to pack into specific lattices more easily, aiding in the formation of protein crystals. Therefore, the removal of flexible regions of protein molecules to create stable protein fragments and domains benefits developing crystallization-favoring crystals [17,18,19]. In situ proteolysis using a trace amount of proteases to diminish flexible or unstable structures is deemed an effective way of improving protein crystallization [20,21,22,23,24]. A target protein sequence can be engineered to express the potential crystallization-favoring truncation forms. On the other hands, proteolysis can be somewhat serendipitous, and the results are often difficult to predict; thus, researchers tend to add various proteases directly to crystallization solutions so as to increase the possibility of crystallizing proteins of interest [24,25,26]. Additionally, digestion decreases the surface conformational entropy of protein surface residues, which is a critical indicator for successful protein crystallization [22,23,27,28].

Chymotrypsin (CT), widely used in in situ proteolysis, primarily targets the non-polar amino acids that contain large aromatic side chains, including tyrosine, tryptophan, and phenylalanine. It accommodates the aromatic groups into its hydrophobic active site and cleaves the carboxyl group of amino acids [29,30,31]. In addition, CT also hydrolyzes the other amide bonds in peptides at slower rates (i.e., leucine and methionine) [32]. CT is usually used for cutting thermally sensitive loops in proteins to stabilize the molecules during crystallization [33,34].

The Type II secretion system (T2SS) is a sophisticated secretion machinery that is utilized by a variety of Gram-negative pathogens to translocate large, structured virulence factors for bacterial infection [35,36]. In *Pseudomonas aeruginosa*, this secretion system consists of twelve Xcp proteins, in which four minor pseudopilins, Xcp-U, -V, -W, and -X, assemble into the tip complex of the piston-like pseudopilus [37], which is critical for the secretion of diverse virulence factors [38,39].

Herein, we report that in situ proteolysis conditions prompted the formation of XcpVWX ternary complex crystals, but, intriguingly, resulted in generating two distinct lattice types in the same crystallization solution under different CT digestion conditions, which are correlated with the CT quantity and digestion duration. With the lower concentration and shorter incubation of CT, complex molecules are packed into a hexagonal lattice with a P3 space group, which contains three XcpVWX complex molecules in one asymmetric unit (ASU). In comparison, upon a longer CT digestion at a higher concentration, the crystals exhibited an P2_1_2_1_2_1_ orthorhombic lattice, with only one ternary complex molecule in each ASU. The orthorhombic lattice is smaller in size than the hexagonal lattice, in which complex molecules are packed more compactly. The averaged atomic displacement parameters (ADP) of the two structures indicate that all three molecules in the hexagonal lattice have a relatively higher structural flexibility than the molecule in the orthorhombic lattice. Therefore, changes in the in situ proteolysis conditions have direct impacts on the molecular packing in crystals, which have implications for improving proteolysis-assisted crystallization, so as to obtain crystals of a better quality.

## 2. Results

### 2.1. Purification of the XcpVWX Complex

The clones of the soluble forms of the individual minor pseudopilins were constructed into pET32a, with N-terminal thioredoxin (Trx) and hexahistidine tags (Figure 1A). A TEV protease cleavage site was engineered in the vector for tag removal. Expressed in *E. coli* codon plus competent cells, all three Trx-tagged proteins (Xcp-V, -W and -X) were individually purified by nickel-NTA affinity chromatography, followed by the cleavage of tags. The purity of tag-free pseudopilins was polished using size exclusion chromatography (SEC). The XcpVWX complex was formed by mixing the three purified pseudopilins. After incubation, the ternary complex was further purified by SEC (Figure 1B). The fractionated complex peak showed a molecular binding ratio of approximately 1:1:1 in the three molecules, according to the SDS-PAGE result (Figure 1C).

### 2.2. Changes In In Situ Proteolysis Conditions Have Led to Distinct XcpVWX Complex Crystal Forms

The SEC-purified XcpVWX ternary complex sample was used for the sitting-drop vapor diffusion crystallization trials. Initial hits of the XcpVWX complex crystals were found in the crystallization solution containing 20% PEG 2000 MME, 0.1 M Tris, pH 8.5, and 0.2 M Trimethylamine N-oxide (TMAO) at 20 °C (Figure 2A,C).

Chunk-like crystals appeared in the starting crystallization condition when the ternary complex was under low-dosage CT digestion (complex = 10 mg/mL, CT = 0.01 mg/mL, complex:CT = 1000:1; Figure 2A,B). Instead of forming single crystals, the crystals under this condition grew into overlapping multi-crystals. Attempts were made to reduce the nuclei in the crystallization drops to produce single crystals, specifically using microseeding. Nevertheless, limited success was seen, as the multi-crystals were difficult to isolate for obtaining usable seeds. During optimization, however, more crystal shapes were observed, including hexagonal, cubic, diamond-like, etc. under the same digestion condition as in the crystallization process (Figure 2B). These crystals usually require three to four days to mature after microseeding.

Interestingly, a 10-fold higher concentration of CT (complex = 10 mg/mL, CT = 0.1 mg/mL, complex:CT = 100:1) produced clusters of needle- or rod-like crystals in a radial pattern (Figure 2C). This type of crystals usually needed relatively longer time (~15 days) after microseeding to develop into a full shape. In contrast to the chunk-shaped crystals, the needle-like crystals tended to feature a more solid shape when experiencing longer digestion (Figure 2D). Despite clustering, single crystals could be isolated from clusters by carefully touching the nucleation centers. These crystals were suitable for X-ray diffraction experiments.

### 2.3. Different Crystal Forms of the XcpVWX Complex Demonstrate Different Crystal Lattices

To better characterize the nature and diversity of the crystal forms of the XcpVWX complex, a high-intensity synchrotron X-ray light source at the Advanced Photon Source was used to acquire diffraction data. The collected diffraction data were indexed and scaled using XDS [40]. Phasing, model building, and structure refinement were carried out using the PHENIX [41] combined with manual refining using Coot [42]. The crystal structure of the XcpVWX complex that originated from the chunk-like crystals was determined (Figure 3A and Table 1).

The chunk-like crystals diffracted the X-rays to the highest resolution of 2.83 Å. The XcpVWX complex crystal belonged to a P3 hexagonal lattice with a cell dimension of *a* = *b* = 158.1 Å and *c* = 64.7 Å. An analysis of the diffraction data by Xtriage confirmed the twinning of the crystals, which was also observed in the crystallization, which contained three merohedral twin operators (Appendix A). The calculated Matthew’s coefficient indicated that there were three molecules in the asymmetric unit (ASU) with a solvent content of 48.3%. In the phasing stage, a stepwise phasing strategy by molecular replacement was implemented to find the three complex molecules in the ASU. The initial round of molecular replacement enabled us to capture the first complex molecule in the unit cell. Treating the first molecule as a partial solution, two further solutions with a similar likelihood gain were computed by Phaser. In terms of the molecule arrangement in the P3 space group, the two solutions reflected the locations of the remaining two ternary complex molecules, separately. Consequently, using either of the two solutions as a partial solution led to building up the entire structure model of the ternary complex (Figure 3A). In the plane formed by the sides of a and b, the molecules were packed relatively loosely to form triangular cavities (Figure 3B, left panel, and Appendix A, left panel). However, it is noted that the packing was tight, based on the side view (Appendix A, left panel), owing to the small c side.

As shown by our previously reported results [43], the long rod-like crystals were obtained via high-concentration CT digestion. The crystals belonged to a P2_1_2_1_2_1_ orthorhombic lattice, with cell dimensions of *a* = 61.54, *b* = 76.76, and *c* = 102.86 Å (Figure 3B, right panel). One ASU contained one XcpVWX complex molecule, with a solvent content of only 34.0%, which is substantially lower than the 48.3% of the hexagonal space group. Unlike the molecular packing in the hexagonal lattice of the chunk-like crystals, in the orthorhombic lattice, molecules were packed into a tighter lattice with symmetry molecules. As a result, limited space was available for the solvent.

To better demonstrate the precise differences between the crystal structures in the two lattices, the complex structure of the orthorhombic lattice was superimposed pairwise with the three individual complex structures in the hexagonal lattice (Figure 4, Appendix A, and Table 2). The overall structures aligned well. The major missing residues in all three structures of the P3 hexagonal lattice were found in the XcpV molecules (Figure 4A, Appendix A, and Table 2). They mainly lacked two connecting loops, namely: (1) 59–66, between the N-terminal α-helix and the β-sheet, and (2) 103–113, between the two β-strands. The β-sheet domains were not well established because of residue missing, especially in XcpV of Mol2. The XcpW structures were all intact in the four structures (Appendix A and Table 2). Some loop regions in XcpX were also missing in the structures of 6UTU (Appendix A and Table 2). However, the loop regions that formed the calcium binding sites in XcpX were well maintained (Figure 4B).

### 2.4. Atom Displacement Parameter Analysis Reveals a Different Flexibility of Molecules in the Two Lattices

The atom displacement parameter (ADP; also called the temperature factor or B-factor) usually reflects the flexibility of the protein structure [44], which is related to the atomic thermal Debye–Waller factor in the lattice dynamics theory [45]. This critical crystallographic parameter is deemed to represent the atomic motion and static displacive disorder in protein structures [44,46].

In the hexagonal P3 lattice, the relatively loose packing in the dimensions of a and b has rendered the residues of all three molecules with considerable flexibility in the lattice, resulting in high ADP values (Figure 5A). Polypeptides colored in red show the highest ADP, indicating a high atomic motion in these regions. Owing to the diversified roles in the associations with the symmetry molecules, the three complex molecules show different ADP distribution patterns throughout the structures. Systematic ADP mapping displays the averaged ADP values of the individual residues across each pseudopilin molecule in the ternary complex (Figure 5B). Certain residues in specific regions score higher than the rest, such as residues 90–100 and 166–176 in XcpW, and residues 83–88, 110–116, and 262–271 in XcpX. Of all of the three complex molecules in the unit cell, Mol1, which mainly accounts for contact formation with symmetry molecules, has a lower overall ADP, ranging from 40 to 100, compared with the other two molecules.

High-ADP regions are found to form in the sides of the triangular cavities between the molecules and adjacent symmetry molecules, which allow for more extensive atomic motions (Appendix A). However, because of the contacts between the interacting molecules, the regions close to the interaction interface are relatively stable (e.g., the assembly bundled by the N-terminal α-helices of the three pseudopilins). The inter-pseudopilin interactions narrow the space and limit the atomic motions of the amino acids (Figure 5A).

However, the average ADP values of the residues are smaller in the P2_1_2_1_2_1_ orthorhombic lattice due to a stronger association between the adjacent molecules (Figure 5C,D). The complex molecules in this lattice are positioned closely to each other, which restrains the atomic motion of the residues. Similar to the ADP distribution pattern in the P3 space group, the interaction interface composed by the N-termini, colored in blue or cyan, has the lowest ADP values (Figure 5C). Despite the relatively higher flexibility in the surface polypeptides, it is evident that in general, ADP values are substantially lower (≤65) than the same regions in the P3 space group (Figure 5D).

Through contact analysis, molecules in the orthorhombic lattice, although lower in molecule numbers, form even more inter-complex contacts with symmetry molecules with regard to the total number of residues involved in the association and the total contact interface area (Appendix A). The establishment of more contacts reduces atomic motions, supporting the low ADP scores in this lattice.

## 3. Discussion

Structural determination and characterization of biomacromolecules have played an indisputably fundamental role in understanding their structures and functions, including target-oriented drug design and development [2,47,48,49]. Structure-guided drug design has remarkably led to developing more specific and efficient therapeutic cures, which facilitates the advent and production of novel drugs in the market. In this regard, researchers are indulged in structurally characterizing protein drug targets to discover potential drug sites for therapeutic purposes [50,51].

As the most powerful and well-established tool, X-ray crystallography has at all times been the predominant methodology used for identifying protein structures. Many optimization strategies are designed and implemented in order to improve the quality of protein crystals for X-ray diffraction. The technology of the in situ proteolysis has been gradually used to improve protein crystallization, especially in cases where proteins are difficult to form crystals. Although the precise mechanisms have not been fully clarified, the method of in situ proteolysis has been proven to be useful in many studies [24,25,52].

In our research, we applied limited chymotrypsin proteolysis to propel the crystallization of the minor pseudopilin ternary complex of XcpVWX. Conspicuously, it is found that the crystal lattice formation and the resulting protein packing are correlated to the quantity of the protease and the duration of digestion used in the crystallization. Dong et al. implied in their paper that the concentration of the digesting enzyme could be varied for exploring optimum conditions [23]. We have illustrated that the quantity of the protease used in the proteolysis induces different ways of packing. Digested by low-concentration CT, the XcpVWX complex molecules pack into a P3 hexagonal lattice, which exhibits large lattice dimensions. Differently, after experiencing a longer CT proteolysis of 100-fold higher in amount, the complex has gone through a more thorough digestion in order to generate a more stable and compact crystal form, generating a stringently packed P2_1_2_1_2_1_ orthorhombic lattice.

Through the analysis of the spatial layouts of the two different lattices, it is noted that along the dimensions of a and b, immense solvent cavities existed between the protein triangles of the same plane in the hexagonal lattice. The inter-plane space allows for an extensive atomic motion of the residues in this crystal form, manifested by the high ADP factors throughout the three complex molecules. The stable orthorhombic form instead has restrained room, which causes low ADP scores for all of the residues in the complex structure. In one specific lattice, the establishment of residue contacts confines the mobility and flexibility of polypeptides, and accordingly, their ADP scores are lower than the residues not involved in contact formation.

From the perspective of the proteolysis-involved crystallization progress at different stages, the complex molecules underwent chymotrypsin digestion while the crystallization and packing were taking place. Aided by digestion, flexible regions that impede packing (i.e., connecting the loops between the secondary structures) were minimized to generate a stable form of molecules. As presented in Figure 4 and Appendix A, some surface unstable loops in different component molecules are found to be missing in the structures, which are highly probable to prevent the association among adjacent molecules from forming a specific lattice. The removal of them reduced the system Gibbs free energy so as to facilitate crystallization. The P3 hexagonal lattice appears to represent a stage of semi-digestion by CT, in which the packing is in a transition state. Insufficient proteolysis, timewise and quantity-wise, refrains the crystal development into a fully packed form. The packing progress would be driven and continue in the presence of more enzymes. This is evident that the two forms were found to coexist in the same drop (Appendix A). It is thus suggested that the in situ digestion-induced molecular packing to generate a specific Bravais lattice is likely dynamic with the progress of proteolysis.

Additionally, based on the crystal structures and the sequences of the pseudopilins, chymotrypsin barely fragmentated the three pseudopilin molecules, but instead removed the flexible regions to stabilize the overall structures, which is not exactly similar to the usage of proteases for protein fragmentation for mass spectrometry.

Interacting contacts formed by protein molecules propel and reinforce lattice packing. Proteolysis may change the interacting behaviors by removing the residues in the flexible regions on an interacting interface between the packing molecules. As proteases have specific recognition sequences, choosing suitable enzymes is critical, because a loss of interaction between molecules or domains can be induced by improper digestion of the interacting residues. A detailed investigation should be conducted for pinpointing appropriate digestion conditions.

Our research has, for the first time, established the correlation between the formation of a Bravais lattice and the conditions of in situ proteolysis. The results reported here demonstrate the impact of proteolysis conditions on molecule packing and crystal formation in the same crystallization solution. Our work suggests that varying proteolysis conditions serve as a useful way not only for optimizing protein crystallization, but also providing the possibility of generating crystals of different lattices, some of which would have a better quality. Taken together, limited in situ proteolysis should be more widely practiced for protein crystallization, particularly in exploring the possibility of obtaining different crystals with a better diffraction quality.

## 4. Materials and Methods

### 4.1. Cloning of the Soluble Form of Minor Pseudopilins

Soluble truncations of individual minor pseudopilins (XcpV (28–129), XcpW (28–237), and XcpX (29–313)) were amplified from *P. aeruginosa* PAO1 genomic DNA using Q5 high-fidelity polymerase (NEB, Ipswitch, MA, USA). Purified DNA inserts were subjected to digestion of Kpn I and Xho I restriction endonucleases (NEB) for 1 h, and introduced into the pET32b vector (Novagen, Darmstadt, Germany), which contains an N-terminal hexahistidine tag and a thioredoxin (Trx) tag, followed by the TEV protease recognition sequence (gaaaacctgtacttccagggt) that was engineered into the vector. The recombinant plasmid was introduced into BL21 (DE3) competent cells following standard protocol for transforming chemically-competent cells. All of the resulting constructs were verified by sequencing.

### 4.2. Expression and Purification of Xcp-V, -W, and -X, and the XcpVWX Complex

Colonies of cells were transferred into 25 mL of an LB medium for overnight culturing. The overnight cultures were inoculated into 500 mL of Terrific Broth for large-scale protein expression. Expression was induced using 1 mM IPTG at 16 °C overnight after the OD_600_ of the culture reached 0.6. The harvested cells were sonicated for lysis using Buffer A (50 mM Tris, pH 8.0, 150 mM NaCl, and 10 mM imidazole), and high-speed centrifugation was used to remove the precipitate and cell debris at 18,000 rpm for 30 min. The supernatant was applied to Ni^2+^-NTA resins for incubation, followed by column washing with 50 mL of Buffer A. The proteins were eluted using Buffer A plus 200 mM imidazole. Eluted Trx-tagged proteins were subjected to overnight TEV protease cleavage together with dialyzing against Buffer B (50 mM Tris, pH 8.0, 150 mM NaCl, and 5 mM imidazole). The digested samples were reloaded to an Ni^2+^-NTA column to remove the Trx tag, and the flow-through that contained the pseudopilins were fractionated. The proteins were concentrated and loaded onto the Superdex 75 column (GE Healthcare) for size exclusion chromatography using Buffer C (25 mM HEPES, pH 7.0, 150 mM NaCl, and 1 mM CaCl_2_). Fractionation was done based onthe UV absorbance of peaks, and the fractions were analyzed by SDS-PAGE for purity. The purified components of Xcp-V, -W, and -X were mixed together at a molar ratio of 1.5:1:1 (XcpV:XcpW:XcpX) and incubated at 4 °C overnight to form the XcpVWX complex. The ternary complex was purified by size exclusion chromatography using Buffer C, and the fractions were assayed by SDS-PAGE.

### 4.3. Crystallization and Structure Determination

The XcpVWX ternary complex was first subjected to in situ proteolysis using chymotrypsin (protein:chymotrypsin = 1000:1, *w*/*w*). Initial chunk-like crystals were discovered in 19%–22% PEG 2000 MME, 0.1 M Tris, pH 8–9, and 0.2 M trimethylamine N-oxide (TMAO) by sitting-drop vapor diffusion at 20 °C. The crystals were optimized by microseeding. Cryo-protectant was 25% of the ethylene glycol in the optimized crystallization solution. The diffraction data were collected at beamline 23-ID-B of the Advanced Photon Source, Argonne National Laboratory (Argonne, IL). Complete data sets containing 360 frames of images were collected at a 1.033-Å wavelength at 100 K from a single crystal with the exposure of 1 s per frame. As XDS is a fast and efficient software package for processing twinned data, it was used to index and scale our datasets [53]. The structure of the XcpVWX complex was phased by molecular replacement (Phaser) using the high-quality crystal structure of the XcpVWX complex (PDB ID:5VTM) as a search model. The computed solution was refined iteratively using PHENIX, which is suitable for handling twinned data [54,55], with manual fitting and refining in Coot.

### 4.4. Calculation of ADP

The overall ADP values of the XcpVWX complex structures were calculated in PyMol, so as to demonstrate the distribution of ADP in each of the ternary complex molecules of the same unit cell in the hexagonal P3 lattice. The ADP analysis of the individual residues of each pseudopilin in the ternary complex was performed using BAVERAGE in the CCP4 program suite [56].

### 4.5. Calculation of the Inter-Molecule Contact Formation

The crystal structures of the XcpVWX complex in the P3 hexagonal lattice (PDB ID: 6UTU) and the P2_1_2_1_2_1_ orthorhombic lattice (PDB ID: 5VTM) were uploaded to the PDBePISA server (https://www.ebi.ac.uk/pdbe/pisa/) for calculating the contact formation between the complex molecules and symmetry complex molecules. The number of residues on both the ASU molecules and the symmetry molecules involved in the contacts and interface area were used for evaluating the contact formation in the two lattice types.

## Figures and Tables

**Figure 1 ijms-21-00308-f001:**
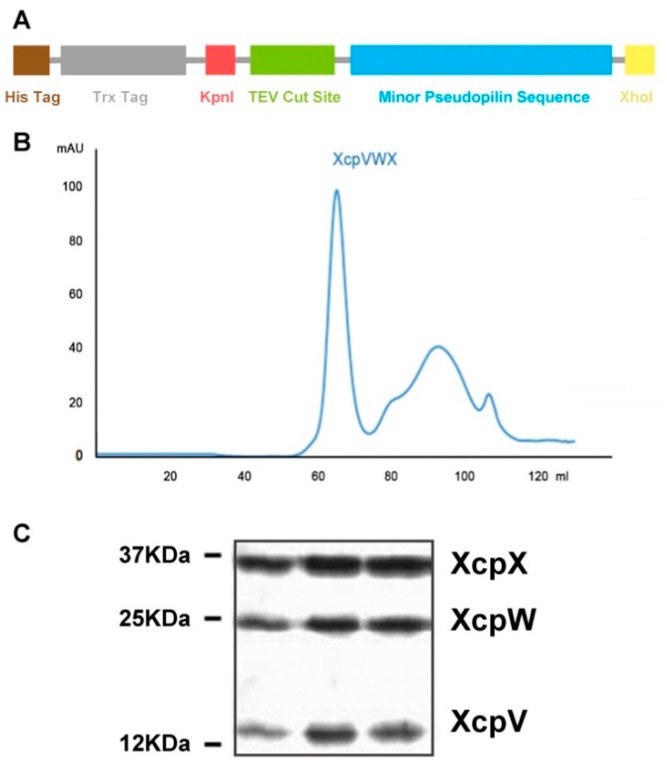
Purification of the XcpVWX ternary complex. (**A**) Schematic of the clone structure of the minor pseudopilins. The sequence of the minor pseudopilin follows the TEV cut site to generate a soluble tag-free form of pseudopilins. (**B**) Purification of the XcpVWX ternary complex through size exclusion chromatography (SEC). The complex peak elutes at 60 mL on a Superdex 75 Hiload gel filtration column. (**C**) SDS-PAGE result of the fractionated peak eluate reveals the formation of the ternary complex by the three pseudopilin molecules.

**Figure 2 ijms-21-00308-f002:**
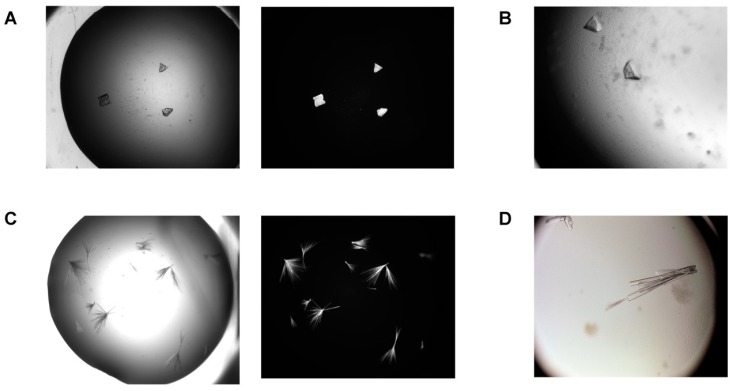
Crystals of the XcpVWX complex before and after optimization. (**A**) Initial hits of the chunk-like, overlapping multi-crystals of the XcpVWX complex under a bright field and UV scopes. The complex was subjected to a low-dosage and shorter digestion of chymotrypsin (CT). (**B**) Optimized chunk-like crystals under bright field. Crystals became larger in size after microseeding. (**C**) Initial crystal hits of the needle clusters of the XcpVWX complex under bright field and UV scopes using high-quantity and longer digestion of CT. (**D**) Optimized needle clusters of crystals under a bright field. This type of crystal not only grew in size, but also formed well-shaped three-dimensional clusters of rod-like crystals.

**Figure 3 ijms-21-00308-f003:**
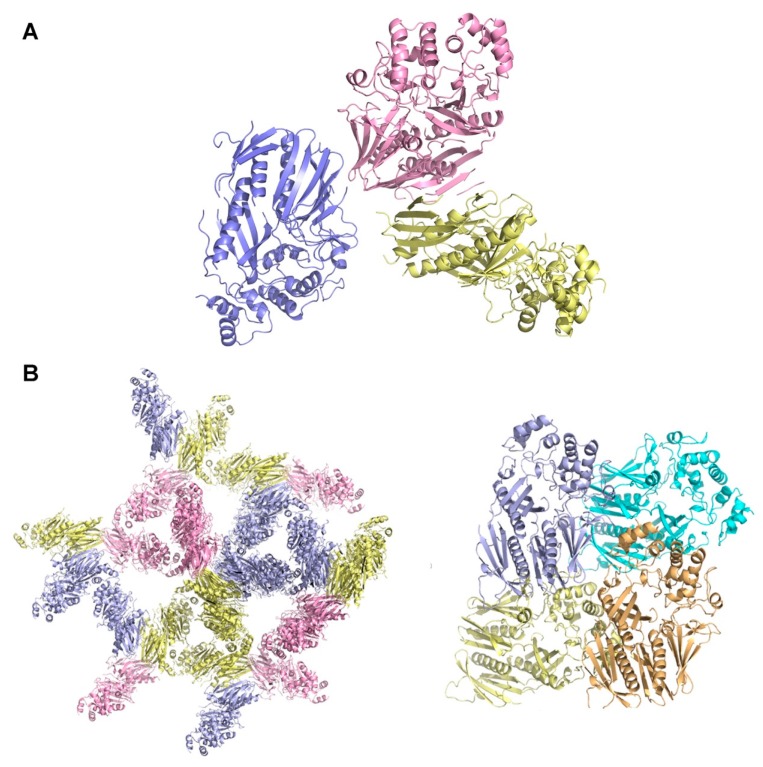
Packing of the complex molecules in the two different lattices. (**A**) Crystal structure of the XcpVWX ternary complex (PDB ID: 6UTU). (**B**) Spatial arrangement of the XcpVWX complex molecules in the P3 hexagonal lattice (left panel) and the P2_1_2_1_2_1_ orthorhombic lattice (right panel). The chuck-like P3 hexagonal crystals show a triangular packing pattern in the lattice. The three molecules in each asymmetric unit (ASU; colored yellow, blue, and pink, respectively) form triangles with the symmetry molecules in the lattice. The compactly packed orthorhombic lattice by XcpVWX molecules is observed in the rod-like crystals (PDB ID: 5VTM). The complex molecules associate with symmetry molecules tightly in this type of lattice. There is only one molecule in each ASU.

**Figure 4 ijms-21-00308-f004:**
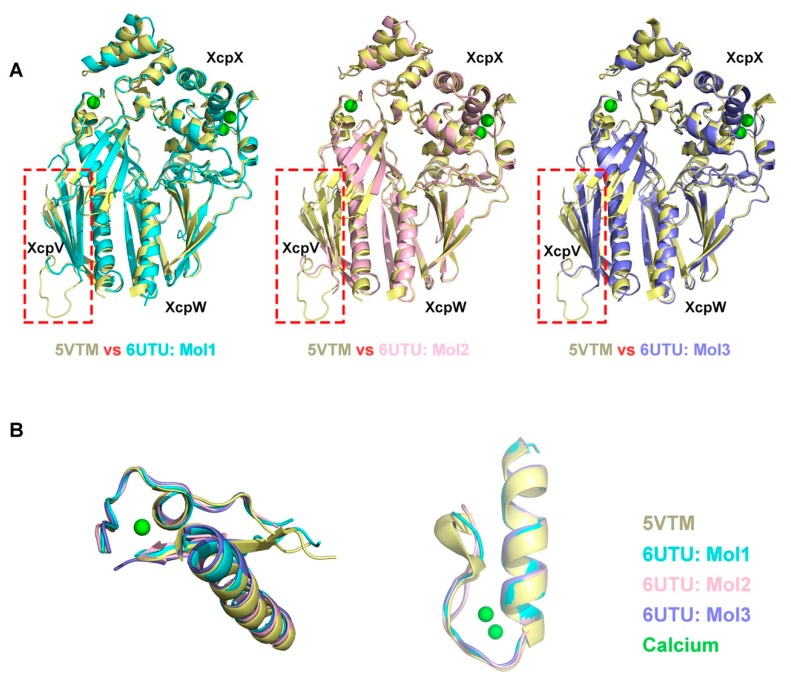
Comparison between the complex structure in the orthorhombic lattice and the ones in the hexagonal lattice. (**A**) The crystal structure of the XcpVWX ternary complex in the orthorhombic lattice (PDB ID: 5VTM) was superimposed onto the individual structures of three complex molecules in the hexagonal lattice (PDB ID: 6UTU). The regions in the dotted box show the main structural differences in pair-wise structure comparisons. (**B**) Calcium binding sites are identical in the four structures.

**Figure 5 ijms-21-00308-f005:**
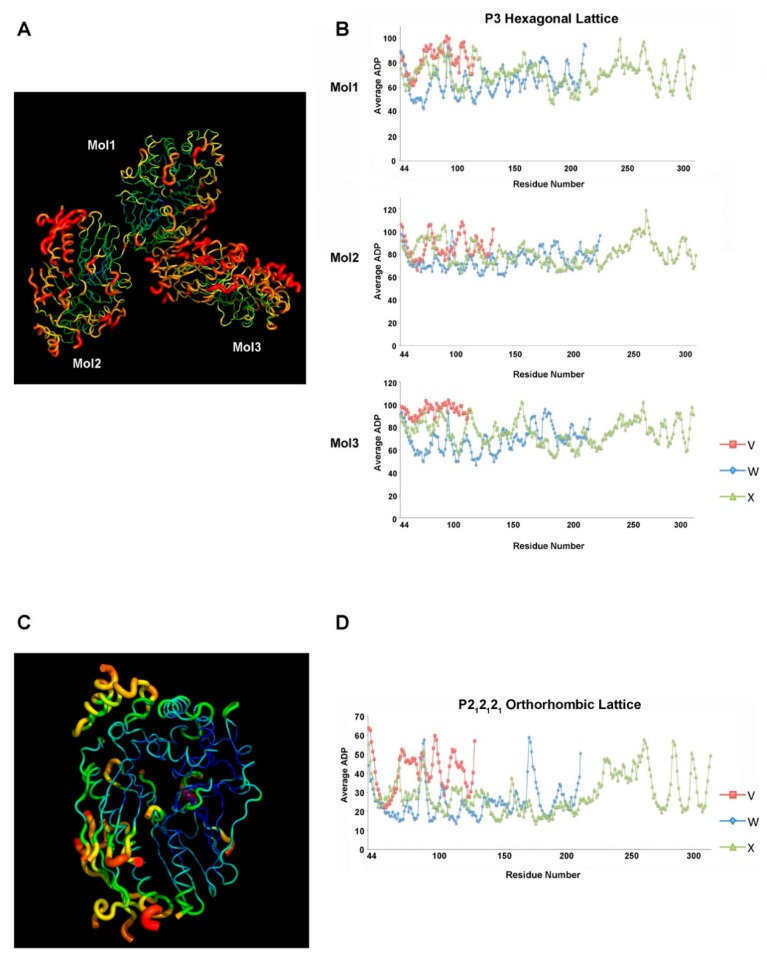
Atomic displacement parameters of the XcpVWX complex in the two types of Bravais lattices. (**A**,**B**) atom displacement parameter (ADP) distribution schematic and ADP values of the protein residues in the hexagonal lattice. (**C**,**D**) ADP distribution schematic and ADP values of protein residues in the orthorhombic lattice.

**Table 1 ijms-21-00308-t001:** Data collection and refinement statistics.

Data Collection	XcpVWX
Space group	*P*3
Cell dimensions	
*a*, *b*, *c* (Å)	158.1, 158.1, 64.7
α, β, γ (°)	120, 120, 60
Resolution (Å)	43.5–2.83 (3.00–2.83) *
Rmerge	13.8 (19.9)
CC (1/2)	99.8 (64.6)
*I*/σ	10.3 (1.6)
Completeness (%)	99.2 (95.7)
Redundancy	10
Refinement	
Resolution (Å)	43.5–2.83
No. unique reflections	42465
*R*_work_/*R*_free_	0.21/0.30
R.m.s. deviations	
Bond lengths (Å)	0.009
Bond angles (°)	1.12

* Values in parentheses are for the highest resolution shell.

**Table 2 ijms-21-00308-t002:** Missing residues in structures.

Protein	Orthorhombic 5VTM	Hexagonal 6UTU		
		Mol1	Mol2	Mol3
XcpV	89–92	60–66 71 79 87–90 104–113	59–65 87–98 103–112	62–65 88–91 106–113
XcpW	87		86	
XcpX	67–75 95 263–267	67–75 93	66–75 92–98 287–291	67–75 91–98 161–163 263–266 286–291

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
