# Peer review of "In Situ Proteolysis Condition-Induced Crystallization of the XcpVWX Complex in Different Lattices"

_ijms, 2020, doi:10.3390/ijms21010308_

Round 1

Reviewer 1 Report

The fact that changing the in situ proteolysis conditions can lead to different crystal forms is an interesting result. However, the authors did not provide any insight into how proteolysis induced the formation of two type of crystals. For instance, which fragment/s were crystallised? this might be check by mass spectrometry.

In table 1, statistics corresponding to the high resolution shell are missing.

Author Response

Dear reviewer,

Thank you very much for your great efforts and valuable suggestions. I would like to express our sincere gratitude to you for the insightful and constructive criticism and comments which have greatly improved quality of the manuscript.

To revise the manuscript, we have carried out additional and detailed analysis of the structures and provided important supporting data, materials and explanations to fully address the your comments and concerns as well as add more contents to clarify our point of view.

I have addressed all comments point-by-point as below (your comments are in italics).

The fact that changing the in situ proteolysis conditions can lead to different crystal forms is an interesting result. However, the authors did not provide any insight into how proteolysis induced the formation of two type of crystals. For instance, which fragment/s were crystallised? This might be check by mass spectrometry.

        We performed the in situ proteolysis by directly adding chymotrypsin into crystallization drops while the complex molecules are crystallizing, which means that digestion happened simultaneously during the crystallization process. This is different than protease treatment and purification of digested protein prior to crystallization. To clarify  this, we have added more contents to elucidate this in Discussion (Line 262-269).

        We really appreciate your great ideas of using mass spectrometry to identify precise post-proteolysis protein sequences. Actually, in our crystal structures we can already observe the missing pieces in proteins through structural analysis. As per your suggestions, we have systematically analyzed the effects of digestion on the crystal structures in Section 2.3 (Line 169-178) by comparing the complex structure in the P212121 orthorhombic lattice with the three complex structures in the P3 hexagonal lattice. New figures (Figure 4 and Figure S2) and table (Table 2) have been implemented to clearly detail the missing residues caused by proteolysis. From structure comparison, although all the molecules have experienced digestion by chymotrypsin, their overall structures are still intact. No protein fragmentation is found  in all the structures. Furthermore, based on the cloned pseudopilin sequences described in MM (Line 295-296), not many residues are lost or unidentified in the structures. Therefore, we have included a paragraph in Discussion (Line 275-278) to clarify this fact.

In table 1, statistics corresponding to the high-resolution shell are missing.

        The data of high-resolution shell have been included in Table 1.

We have also corrected typos, improved the language used in the manuscript and polished the entire context according to your suggestions. 

Thank you again for your help and look forward to seeing your further advice.

Reviewer 2 Report

In this paper, the researchers conducted in situ proteolysis in the crystallization of the XcpVWX using chymotrypsin (CT) digestion. They demonstrated that different CT concentrations and digestion time can result in distinct crystal packing patterns in the same crystallization solution. Overall the paper is well organized with sufficient data presented. However, a few things will still need to be addressed in the discussion section.

Since different proteolysis conditions will give different crystal packing patterns, how to choose the right condition in further optimization procedure? It seems that in different lattices, the interaction between different domains has been changed. How do you know which condition is the right one if the interaction study between domains is important?

Author Response

Dear reviewer,

Thank you very much for your great efforts and valuable suggestions. I would like to express our sincere gratitude to you for the insightful and constructive criticism and comments which have greatly improved quality of the manuscript.

To revise the manuscript, we have carried out additional and detailed analysis of the structures and provided important supporting data, materials and explanations to fully address the your comments and concerns as well as add more contents to clarify our point of view.

I have addressed all comments point-by-point as below (your comments are in italics).

1. In this paper, the researchers conducted in situ proteolysis in the crystallization of the XcpVWX using chymotrypsin (CT) digestion. They demonstrated that different CT concentrations and digestion time can result in distinct crystal packing patterns in the same crystallization solution. Overall the paper is well organized with sufficient data presented. However, a few things will still need to be addressed in the discussion section.

We have added more contents to the discussion section to explain the effects of chymotrypsin-involved digestion on crystallization to rationalize the in situ proteolysis in crystallization. 

We have also included more detailed discussion in Section 2.3 to systematically compare the complex structure in the P212121 orthorhombic lattice with three complex molecules in the P3 hexagonal lattice. New figures (Figure 4 and Figure S2) and table (Table 2) have also clearly detailed the missing residues caused by proteolysis.

2. Since different proteolysis conditions will give different crystal packing patterns, how to choose the right condition in further optimization procedure? It seems that in different lattices, the interaction between different domains has been changed. How do you know which condition is the right one if the interaction study between domains is important? 

Your point of view on this aspect is enlightening. Accordingly, we have expanded our discussion to address this point (Line 279-284).

Thank you again for your help and look forward to your further advice.

Round 2

Reviewer 1 Report

The hexagonal crystal form is twinned according to the XTRIAGE analysis.

Did the authors refine the structure considering twinning? Which twinning law was used?

Author Response

Dear Editor/Reviewer,

Thank you very much for your further comments on the manuscript. We have updated necessary contents based on your question.  Further language improvement has also been done.

The hexagonal crystal form is twinned according to the XTRIAGE analysis. Did the authors refine the structure considering twinning? Which twinning law was used?

             Our crystals are multi-crystals which contain twinning operator(s). The diffraction pattern of this type of crystals showed overlapping spots. Therefore, we used XDS to index and scale data since this software package is considered to be the most efficient method (Ref. 53). We have included this paper in Reference. Results of Xtridge analysis indicated twinned data and  calculated the possible twin law and operators. As per your request, we have included the relevant information in Supplementary Materials (Line 138 and Table S1).

             The reason of using phenix.refine is that it has demonstrated the ability of dealing with twinned data (Ref. 54 and 55). We used the program with proper settings to refine the twinned data following the methods described in the reference. 

             We have also included description of twinned data processing in MM (Line 329-333).

Sincerely yours,

Zongchao Jia